# Noninvasive Flow Monitoring in Simple Flow Phantom Using Resistive Strain Sensors

**DOI:** 10.3390/s21062201

**Published:** 2021-03-21

**Authors:** Sunghun Jung, Dae Yu Kim

**Affiliations:** 1Department of Electrical Engineering, Inha University, Incheon 22212, Korea; tjdgns3481@naver.com; 2Department of Electrical Engineering, Inha Research Institute for Aerospace Medicine, and Center for Sensor Systems, Inha University, Incheon 22212, Korea

**Keywords:** flow monitoring, flow parameters, noninvasive measurement, pulsatile flow, strain sensor

## Abstract

In this paper, we introduce a monitoring method for flow expansion and contraction in a simple flow phantom based on electrical resistance changes in an epidermal strain sensor attached to the phantom. The flow phantom was fabricated to have a nonflat surface and small modulus that are analogous to human skin. The epidermal sensors made of polydopamine and polyvinyl alcohol show sufficient linearity (R = 0.9969), reproducibility, and self-adhesion properties, as well as high sensitivity to small modulus measurements (<1% tensile strain). Pulsatile flow monitoring experiments were performed by placing the epidermal sensor on the flow phantom and measuring the relative changes in resistance by the heartbeat. Experiments were conducted for three types of vessel diameters (1.5, 2, and 3 mm). In each of the experiments, the vessels were divided into Top, Middle, and Bottom positions. Experiments for each position show that the relative changes in resistance increase proportionally with the diameter of the vessel. The vessels located close to the epidermal layer have greater relative electrical changes. The results were analyzed using the Bernoulli equation and hoop stress formula. This study demonstrates the feasibility of a noninvasive flow monitoring method using a novel resistive strain sensor.

## 1. Introduction

In hemodynamics, the Hagen–Poiseuille equation, Navier–Stokes equation, and Bernoulli’s law explain the relationships among vessel diameter, vessel resistance, and pressure difference. The Poiseuille equation shows that the resistance of the blood vessel has an inversely proportional relationship with the fourth power of the vessel diameter [1]. In the Navier–Stokes equation and Bernoulli’s law, the pressure differences in a given streamline are numerically explained as being proportional to the blood vessel diameters [2]. These laws demonstrate that vascular information, including vessel resistance, blood flow velocity, and blood pressure differences, can be predicted by monitoring the changes in the vessel diameters.

Conventional methods for monitoring blood flow include Doppler flowmetry [3,4], spectroscopy [5,6], sonography [7,8], computed tomography [9], electrical capacitance [10], photon attenuation [11], and ultrasound [12], which are used to visualize or analyze vessel structures as well as fluid flow. These methods are commonly used in hospitals as the instruments are expensive, have complicated operations, and require trained operators for accurate analyses [13]. Recently, researchers have proposed implantable sensors that can be placed over the blood vessels to continuously monitor the vessel diameters [13] or blood flow [14,15]. The implantable sensor involves catheter insertion directly into the vessel [16] or wrapping around the blood vessel [17,18,19] to obtain the vessel structure. This sensor can obtain information about the vessel structure and blood flow continuously. These methods should generally be used only on patients who have received vascular surgery or another form of surgery for sensor implantation; moreover, additional surgical procedures may be required to remove these sensors [13,14,15,16,17]. However, none of these methods address general measurements exclusive of the patients.

During the past decade, strain sensors have been widely used to measure blood pressure, blood flow, and pulse waves [13,18,20,21,22,23]. Table 1 compares the relative merits and characteristics of each of these techniques. Researchers place the strain sensors on the epidermis and measure the subtle deformations of the skin caused by cardiac output [20,21]. Resistive strain sensors have suitable characteristics, including a simple fabrication process, a high gauge factor, and an abundant material source for biocompatibility, for epidermal applications [24]. The method of measuring cardiovascular function using the strain sensor is critical to continuous vessel monitoring but is yet to be investigated in depth.

Herein, we propose a method for monitoring vasodilation and vasoconstriction by measuring the electrical resistances using epidermal strain sensors attached to a simple flow phantom. We measured the pressure changes transmitted to the skin through contraction and expansion of the flow phantom with polydopamine (PDA) strain sensors [25] and measured the changes in resistance values depending on the diameter and position (depth from epidermis) of the vessels. The originality of our study is that the developed strain sensor measures tension signals by pressure changes as well as enables flow monitoring noninvasively. In addition, we not only demonstrate the electrical characteristics of the strain sensor, including reproducibility and response time, but also acquire the relative electrical resistance changes during pulsatile flows for tubes with different diameters using a simple flow phantom model. Using this methodology, we demonstrate that there is potential for continuous, efficient, and noninvasive monitoring of flow changes in vivo. 

## 2. Numerical Principle of Measurement

Figure 1 shows the schematic for the vasodilation and vasoconstriction measurement method using the epidermal strain sensors. The fluid flow in the vessels inside the phantom causes structural changes in the vessels. The pressure transmitted to the skin phantom by these structural changes in the vessels can be approximated by Equation (1):(1)$$P_{Total}=P_{out}-P_{phantom}$$
where $P_{Total}$ is the pressure transmitted to the skin phantom, $P_{out}$ is the pressure produced in the vessel by the fluid, and $P_{phantom}$ is the pressure exerted by the phantom on the vessels. $P_{Total}$ causes strain sensor deformation, inducing changes in electrical resistance. Each layer of the skin and vessels in the simple flow phantom have similar mechanical properties to those of human skin and vessels [26,27,28,29]. Thus, $P_{phantom}$ can be obtained through the phantom thickness of each layer and the material composition ratio of the phantom.

$P_{out}$ continuously changes with the flow. In a cylindrical vessel with laminar flow, a Newtonian liquid is used to calculate the pressure difference and flow according to changes in the vessel diameter [30]. To observe the linear relationship between the flow and pressure difference, water is used as the representative Newtonian fluid as it is the most attractive option for producing pulsatile fluid flow in the phantom [31]. Using a syringe pump, a periodic pulsatile fluid flow is established in the vessels. The pulsatile flow induces changes in the diameter and creates pressure differences in the vessels. The Poiseuille equation can be used to expresses this relation as [1,32]
(2)$$Q=\frac{\pi d^{4}\Delta p}{128\mu L}$$
where d is the vessel diameter, Δp is the pressure difference between the two ends of the vessel, μ is the dynamic viscosity, and L is the length of the vessel. Thus, the changes in flow control the contractions and expansions of the vessels inside the phantom. 

To describe the fluid dynamics of incompressible and Newtonian fluids flowing within the phantom, the Navier–Stokes equation is used as follows:(3)$$\rho \left[\frac{\partial v}{\partial t}+(v\cdot \nabla )v\right]=-\nabla p+\rho g+\mu \nabla^{2}v$$
where ρ is the density of the fluid, v is the mean fluid velocity vector, p is the dynamic pressure, μ is the dynamic viscosity, and g is the gravitational acceleration. These terms indicate that the acceleration of a fluid depends on the pressure (−∇p), volume force (ρg), and viscous shear stress ($\mu \nabla^{2}v$). The Navier–Stokes equation numerically expresses the conservation of mass, momentum, and energy in the dynamic fluid. This equation can be simplified using Bernoulli’s law by excluding viscosity from the pressure gradient and flow rate. Figure 2 shows the variations in flow velocity and pressure depending on the diameter of the vessel for a given streamline. By Bernoulli’s law, the sum of all forms of energy in the streamline for a flowing fluid is constant and can be expressed as
(4)$$P_{1}+\frac{1}{2}\rho v_{1}{}^{2}+\rho gh_{1}=P_{2}+\frac{1}{2}\rho v_{2}{}^{2}+\rho gh_{2}$$
where $P_{1}$, $P_{2}$ are the pressures at two points in the vessel with corresponding flow rates $v_{1}$, $v_{2}$, and heights $h_{1}$, $h_{2}$ from a common horizontal ground plane, respectively. Equation (4) can now be expressed as [33]
(5)$$P+\frac{1}{2}\rho v^{2}=\mathrm{constant}$$
because $\rho gh_{1}$ and $\rho gh_{2}$ along any particular streamline are negligible. Thus, as the vessel diameter increases, the cross-sectional area of the vessel increases, and the flow rate decreases. Owing to the decreased flow velocity, the kinetic energy ($\frac{1}{2}\rho v^{2}$) decreases and pressure increases to maintain constant overall energy. Therefore, $P_{out}$ continues to change as the diameter of the vessel changes.

When the pressure injected into the vessel is equal, $P_{out}$ is explained by the hoop stress of the vessel. If the wall thickness of the vessel is no more than one-twentieth of its diameter, the thin-walled assumption is valid. In the thin-walled cylindrical tube, for an injected fluid flow, pressure is generated and transmitted to the wall of the tube; this is called the hoop stress of the tube and is calculated using the following equation [34]:(6)$$\sigma_{\theta }=\frac{p_{i}D}{2t}$$
where $\sigma_{\theta }$ is the hoop stress, D is the tube diameter, $p_{i}$ is the pressure of the fluid flow, and t is the wall thickness. Thus, the diameter of the vessel, pressure, and wall thickness determine the hoop stress [35]. If the same $p_{i}$ is applied to other vessels with the same wall thicknesses, for a larger diameter, the hoop stress is greater [35]. Therefore, the vessel diameter can be predicted by comparing the hoop stresses at a given pressure.

## 3. Methods and Experimental Setup

### 3.1. Simple Flow Phantom

The structural and biophysical characteristics of human skin can be defined by its acoustic, optical, and mechanical properties. Because the epidermal sensor operation is based on the mechanical properties of the phantom, the fabrication of phantoms, including all of their characteristics, is excessive for monitoring vessels with the epidermal sensor. In this study, we fabricated three simple flow phantoms with vessel diameters of 1.5 mm, 2 mm, and 3 mm by considering the mechanical properties. Each flow phantom included polyvinyl alcohol (PVA) tubes [36] at different distances from the bottom of the phantom, as shown in Figure 3: 17 mm (Bottom), 19 mm (Middle), and 23 mm (Bottom). For adults, the vessels on the face are usually 1–3.6 mm in diameter [13,37], carotid vessels are mainly 2–3 mm in diameter [38], and blood vessels on the arms are less than 4 mm in diameter [39,40]; thus, we choose the diameter of the representative vessels in this experiment as 1.5 mm, 2 mm, and 3 mm, respectively. The tube thickness is approximately 0.1 mm and is constant in all vessels. Figure 3 shows the overall diagram of the phantom. The flow phantom consists of four layers: base layer, hypodermis, dermis, and epidermis [26]. Table 2 summarizes the composition of the flow phantom for each layer. In addition, Table 3 shows the Young’s modulus for each layer of the phantom and human skin.

First, to prepare the simple flow phantom, we used an acrylic container to stack each layer. The inner dimensions of the acrylic container were 32 mm × 120 mm × 35 mm (height × width × depth). Vessel-sized holes were then drilled at 30 mm intervals on the longer wall of the acrylic container at heights of 17 mm, 19 mm, and 23 mm from the bottom of the container (i.e., Bottom, Middle, and Top positions, respectively). The base layer was composed of 8 wt% gelatin (G1890-Type A, Sigma-Aldrich, St. Louis, MO, USA), 1 wt% 1,2-hexanediol (213691, Sigma-Aldrich, St. Louis, MO, USA), and deionized distilled water (DW). The base layer solution was mixed for 2 h at 2000 RPM using a homogenizer at 60 °C DW. The homogeneously mixed base layer solution was stacked in the acrylic container to a thickness of 15 mm. The container was then sealed to prevent air bubbles in the solution and allowed to cool to ambient temperatures in the range of 26 °C to 27 °C. When the base layer was fully gelled, the vessel phantoms (Single layer type, WetLab, Japan) connected to the syringe dispenser were inserted into the holes of the acrylic container and fixed. The vessel phantoms were placed in the hypodermis layer, which was composed of 2 wt% gelatin, 0.2 wt% agar (A1296, Sigma-Aldrich, St. Louis, MO, USA), 1 wt% 1,2-hexanediol, and DW, and the solution was mixed for 2 h at 2000 RPM using the homogenizer at 60 °C DW. The solution was then sealed and allowed to cool to 40 °C. Before the solution gelled completely, the solution was poured over the base layer to a thickness of 10 mm to submerge the vessel phantom. The dermis layer was composed of 24 wt% gelatin, 1 wt% agar, 1 wt% 1,2-hexanediol, and DW, and this solution was mixed for 2 h at 3000 RPM using a homogenizer at 60 °C DW. The prepared solution was cooled to 40 °C and poured over the hypodermis to ensure that the dermis layer had a thickness of 1 mm. Finally, the epidermis layer was prepared with 0.1 g/mL of gelatin, 0.05 g/mL of glycerol (49781, Sigma-Aldrich, St. Louis, MO, USA), 0.1 wt% of 1,2-hexanediol, and DW, and the solution was spread over the dermis layer to a thickness of 0.1 mm.

### 3.2. Epidermal Strain Sensor

The pulsatile fluid in the vessel generates pressure, which is then transmitted to the epidermal layer. The goal of the epidermal strain sensor is to measure the microstrain (<1% tensile strain) delivered by the pressure to the epidermal layer. For accurate measurement, the epidermal sensor was attached directly and without separation to the phantom because the epidermis has a nonflat surface. We used epidermal strain sensors with self-adhesive patches that could detect microstrains transmitted to the skin. The epidermal sensors consist of polydopamine (PDA) and polyvinyl alcohol (PVA; 341584, Sigma-Aldrich, St. Louis, MO, USA) [25]. The use of PDA, a mussel-inspired substance, promotes uniform distribution within a hydrogel and allows excellent adhesion to various surfaces regardless of the chemical properties of the material [41]. The sensor using PDA has self-adhesion characteristics, which attaches well to human as well as porcine skin (gelatin) [25,41].

Figure 4 depicts the entire process of sensor fabrication. Dopamine forms PDA chains by polymerizing in the alkaline condition. Dopamine hydrochloride (DA; H8502, Sigma-Aldrich, St. Louis, MO, USA) was added to the buffer solution (109460, Sigma-Aldrich, St. Louis, MO, USA) at a pH of 8 to achieve polymerization at 60 °C for 24 h to prepare PDA. The DA to buffer solution mass ratio is 15 (DA/buffer solution = 15 wt%). After the polymerization process, the PDA is cooled to ambient temperature. The PVA solution is a mixture of PVA powder and DW at a ratio of 10 wt%. The homogenizer stirs the PVA solution at 2000 RPM at 95 °C for 2 h. When the PVA solution forms an opaque gel, it is cooled to ambient temperature. The sodium tetraborate solution (Borax; 221732, 99% purity, Sigma-Aldrich, St. Louis, MO, USA) used here is a mixture of sodium tetraborate and DW at a ratio of 8 wt%. The homogenizer stirs the Borax at 3000 RPM at 60 °C for 2 h. The prepared PVA and PDA solutions are then mixed in a mass ratio of 1:9. The Borax is then slowly poured into the homogenously mixed solution of PDA/PVA at a mass ratio of 6:1 (PDA/PVA: Borax = 6:1). When the solution is mixed, it changes to gel form. To remove trapped air from the gel, we sealed the gel in a glass bottle and ultrasonicated the mixture for 4 h. When the process of ultrasonication is complete, the hydrogel of the epidermal sensor is obtained.

Figure 4e presents a schematic of the encapsulation process for the epidermal strain sensor. The gel-type sensor is encapsulated within acrylic tape (VHB-4910, 3M, Maplewood, MN, USA) to measure the sensor characteristics, including linearity, response time, and reproducibility. First, the acrylic tape is prepared by cutting strips of 1 mm × 70 mm × 15 mm (height × width × depth). Then, the copper tape (1181, 3M, Maplewood, MN, USA) is attached to the acrylic tape and connected to 8 AWG wires. The acrylic tape is placed over 1 mm × 70 mm × 5 mm (height × width × depth) areas on both sides of the acrylic tape to form a groove. The epidermal sensor is then inserted into this groove. Lastly, the top with the acrylic tape is covered and encapsulated. The tensile force of each encapsulated sensor is evaluated with a universal testing machine (UTM; 5569, Instron, Norwood, MA, USA). At the same time as the tensile force experiment of the sensor, data on variation of the electrical resistance according to the strain of the sensor are obtained using a source meter (Keithley 2400, Tektronix, Beaverton, OR, USA).

### 3.3. Experiments of Pulsatile Flow

Syringes were connected to the vessels through dispensing needles of the same sizes as the vessel diameters. Because air bubbles in the syringe produce turbulent flow and cause errors in the test results, the water in the syringes was prepared without bubbles. Water maintains the temperature of the system because viscosity changes with temperature. We regularly changed the pumping rates between 6.25 mL/min and 10.25 mL/min using a programmable syringe pump (NE 4000, New Era Inc., Farmingdale, NY, USA) to create fluid flow. An injected flow at a rate of 10.25 mL/min extends the diameter of the elastic vessel according to the Poiseuille equation, and an injected flow at a rate of 6.25 mL/min causes the diameter of the vessel to contract. As a result, the vessels experience dynamics analogous to vasodilation and vasoconstriction. The pressure generated by periodic expansion and contraction of the vessel is transferred to the epidermal layer, which causes deformation of this layer periodically. According to Bernoulli’s law, the pressure that the vessel exerts on the skin depends on the diameter of the vessel. To determine how the diameter of a vessel affects the exerted pressure on the skin, we conducted experiments using different vessel diameters. Each experiment was maintained at the same ambient temperature, water temperature, and flow rate. To maintain the water temperature, we placed water on a hot plate (HSD150-03P, 4science, Pangyo, Seongnam, KOREA) at a setting of 60 °C. The experiments were repeated for each diameter. The copper tape and electrical wires were connected to the container wall of the phantom and source meter. The gel-type strain sensor was placed on the epidermis and connected to the copper tape. The changing pressure of the epidermal layer induced strain sensor deformations and caused changes in its electrical resistance. The source meter acquired these electrical resistance variations from the deformations of the strain sensor. LabVIEW (National Instruments, Austin, TX, USA) software installed on a laptop was used to acquire and monitor the resistance change data from the source meter. The acquired resistance data were then plotted as graphs using the graphing program Origin (OriginLab Corporation, Northampton, MA, USA).

## 4. Results

### 4.1. Sensor Characteristics

The encapsulated sensor was first connected to the universal testing machine (UTM; 5569, Instron, Norwood, MA, USA). To monitor linear changes in electrical resistance due to increase in the tensile strain, the UTM applied a tensile strain to the epidermal sensor at a rate of 6 mm/min. Two wires from the sensor were connected to the source meter; the electrical resistance was measured by continuously applying a bias voltage of 5 V to the sensor. Figure 5a shows the increase in tensile strain over time and the resulting changes in the electrical resistance. The graph shows that the electrical resistance increases linearly with strain (R = 0.9969). The electrical resistance changed from 2.62 kΩ to 48 kΩ when the tensile strain increased from 0% to 450%.

The skin has a small elastic modulus of 140 kPa to 600 kPa when the epidermal thickness is 0.05 mm to 1.5 mm and 2 kPa to 80 kPa when the dermal thickness is 0.3 to 3 mm [42]. The small modulus of the skin induces small deformations in the sensor attached to the skin. Generally, sensors are highly sensitive to small deformations. A tensile strain of 0.1% to 0.5% was applied to the sensor to observe its responses to small strains. The resistance was measured by increasing the tensile strain by 0.1%. The strain was increased and held for 10 s to confirm that the sensor maintained the resistance for a given strain. As the tensile strain increased to 0.5%, the resistance of the epidermal sensor was measured by reducing the strain by 0.1% (step-keep change test) [25]. Figure 5b is a graph of the sensor resistance values for the strain changes and maintenance. The epidermal strain sensor tends to show increased resistance at a constant rate when the same strain is applied, but the electrical resistance tends to increase linearly when the strain is maintained. When the electrical resistance is within the first 0.1% of strain and returned to 0.1% from 0.5% strain, there is an error of about 0.01%. The calculated gauge factor (ΔR/R)/(Δl/l) of the developed sensor is 38.93 at 0.5% tensile strain, which exhibits enough sensitivity for monitoring pressure changes on the skin phantom. The gauge factor of our proposed device is higher than the sensitivity of the reference article [43] using carbon nanotubes.

The immediate response of the sensor is one of the important parameters in the operation of the strain sensor [44]. The vessels continuously experience changes in their volumes. These changing volumes induce deformation of the sensor. The immediate changes in resistance resulting from deformation allow real-time monitoring of the vascular information. Figure 5c shows the response times of the epidermal sensor graphically. When 0.1% strain is applied to the sensor, the response time is 96 ms. 

### 4.2. Fluid Flow Monitoring

The vessel is placed at the Top, Middle, and Bottom positions for each vessel’s diameter to monitor the sensor responses according to the diameter and position of the vessel. The fluid flow produced by the syringe pump is transmitted to the vessels in the Bottom, Middle, and Top positions of the flow phantom. Experiments were conducted with three different vessel diameters, and pulsations were simulated by modifying the flow rate of the syringe pump. When the flow was 6.25 mL/min, the resistance of the strain sensor was $R_{0}$. Here, ΔR represents the difference between the resistances of the sensor when the fluid flows are 10.25 mL/min and 6.25 mL/min. The time between pulses is approximately 3.32 s. Because the length of the vessel and the viscosity of water are constant, changing the fluid flow from 6.25 mL/min to 10.25 mL/min increases the diameter of the vessel according to Equation (2). Equation (5) describes the increase in the diameter of the vessel for a given streamline, indicating a decrease in fluid flow velocity and increase in pressure. Because the thickness of the phantom pressing on the vessel depends on the position of the vessel, the higher the height difference between the Bottom and Top positions, the smaller is the value of $P_{phantom}$ in Equation (1). The $P_{out}$ is the same and $P_{phantom}$ becomes smaller, thereby increasing $P_{Total}$ in Equation (1) for the same diameter of the vessel.

We generated four pulses by operating the syringe pump. For each pulse, the relative electrical resistance changes were measured as along with the means and standard deviations to calculate the four measurements. Figure 6a shows the changes in electrical resistance caused by pulsatile fluid flow according to the position of vessels for the 1.5 mm diameter. Figure 6b shows the average of the relative rate of change in electrical resistance depending on the position of the vessel. 

The 1.5 mm vessels show average relative electrical resistance changes of 0.03%, 0.07%, and 0.17% at the Bottom, Middle, and Top positions, with standard deviations of 0.0029%, 0.0052%, and 0.0051%, respectively. Compared to the Bottom position, the mean of the relative change in electrical resistance differed by 2.37 and 5.29 times for the other two positions.

Figure 7a shows the changes in electrical resistance caused by pulsatile fluid flow according to the position of the vessels of 2 mm diameter. The time between pulses is approximately 3.32 s. When pressure is applied to the vessel with a constant thickness, the increase in the diameter of the vessel indicates an increase in the pressure transmitted to the skin, as explained by Equation (6). Figure 7b shows the average relative rate of change in electrical resistance depending on the position of the vessel.

The 2 mm vessels have average relative electrical resistance changes of 0.05%, 0.11%, and 0.21% at the Bottom, Middle, and Top positions, with standard deviations of 0.0034%, 0.0069%, and 0.0086%, respectively. Compared to the Bottom position, the mean relative change in electrical resistance differed by 2.16 times in the Middle and 4.09 times in the Top positions. The relative changes in the electrical resistance are 1.66, 1.51, and 1.28 times greater than those of the vessel with 1.5 mm diameter for the Bottom, Middle, and Top positions, respectively.

Figure 8a shows the changes in electrical resistance caused by pulsatile fluid flow according to the positions of vessels with 3 mm diameter. The time between pulses is approximately 3.36 s. The 3 mm vessels have average relative electrical resistance changes of 0.10%, 0.20%, and 0.34%, at the Bottom, Middle, and Top positions, with standard deviations of 0.0089%, 0.0081%, and 0.0139%, respectively. Figure 8b is a graph showing the average relative rate of change in electrical resistance depending on the position of the vessel. 

Compared to the Bottom position, the mean relative changes in electrical resistance differed by 2.05 and 3.39 times for the other two positions. The relative changes in electrical resistance were 3.17, 2.74, and 2.03 times greater than those of the 1.5 mm vessel for the Bottom, Middle, and Top positions, respectively. Compared to vessels with 2 mm diameter, the changes in the relative electrical resistances were 1.91, 1.81, and 1.58 times for the Bottom, Middle, and Top positions, respectively.

## 5. Discussion

The goal of our study was to formulate a new method for vessel monitoring and to demonstrate the validity of the proposed method by experiments using simple flow phantoms. We propose this noninvasive method instead of conventional vessel monitoring, which requires expert analysis [3,4,5,6,7,8,9,10,11] or invasive methods [16,17,18,19]. The movement of the vessels cause pressure variations that are transmitted to the skin to produce subtle changes. We measured and analyzed these subtle changes to monitor the vessel movements. We noted that the fabricated strain sensor could measure subtle deformations on the skin. The strain sensor has the characteristic that its resistance changes when strain is applied simply. We then analyze the sensor data mathematically to understand the mechanism of vascular motion and to estimate the relative diameters and positions of the vessels.

According to the Navier–Stokes and Bernoulli equations, the amount of fluid in a streamline remains the same and moves [33]. When the flow was maintained, we demonstrate from Figure 6a, 7a, and 8a that the closer the vessel is to the epidermal layer, the greater is the pressure produced. This result is the effect of skin thickness that presses upon the vessels. Comparing the average value of each relative resistance change in Figure 6b, 7b, and 8b, we see that the larger the diameter of the vessel at a location, the greater is the pressure applied to the skin. As the position of the vessel approaches the epidermis and the effect of thickness of the skin on the vessel decreases, the growth rate of the diameter of the vessel and increase in pressure exerted by the vessel toward the skin change similarly. For example, when the vessel of 1.5 mm diameter was changed to 2 mm diameter, the diameter increased by 1.33 times, and for a closer Top position, the relative resistance change to 1.33 was also closer. This means that as the effect of the skin decreases, the pressure on the strain sensor becomes equivalent to the pressure produced by the vessel itself, which can be explained by hoop stress [35].

This method has some limitations. First, there is an inherent limitation with respect to the high sensitivity of the epidermal strain sensor. In other words, high resistivity changes by external interference have not been solved when detecting microstrains. Second, there are several clinical studies on wearable applications of strain sensors but not many of these studies are targeted at hemodynamics. This causes difficulties in comparing experimental and clinical data; further, mathematical hypotheses and validations are required. Third, the effects of bone, sweat glands, viscosity of the actual blood, and vessel stiffness [45] on the skin are not considered in this study because of complexities and difficulties in fabricating a real skin–vessel phantom. In the future, our research will focus on these issues to improve the reliability of the vessel monitoring method for clinical applications. 

## 6. Conclusions

The objective of this study was to formulate a noninvasive method of monitoring vasodilation and vasoconstriction from skin deformations. A strain sensor was fabricated and used to monitor and analyze the vessel dynamics in the skin indirectly. The validity of the method was experimentally demonstrated using a simple flow phantom, which was fabricated with three different vessel diameters (1.5 mm, 2 mm, and 3 mm) and divided into Top, Middle, and Bottom vessel placements. We observed that pulsatile flows in the vessels of 1.5 mm, 2 mm, and 3 mm diameters caused relative changes in the electrical resistance ranging from 0.03% to 0.33%. The vessels close to the Top position had similar growth rates for vessel diameter and relative resistance changes. When the diameter of the 1.5 mm vessel was doubled, the change in electrical resistance at the Top position was 2.03 times. The proposed method thus has the potential to be a new approach for monitoring hemodynamics in the human body.

## Figures and Tables

**Figure 1 sensors-21-02201-f001:**
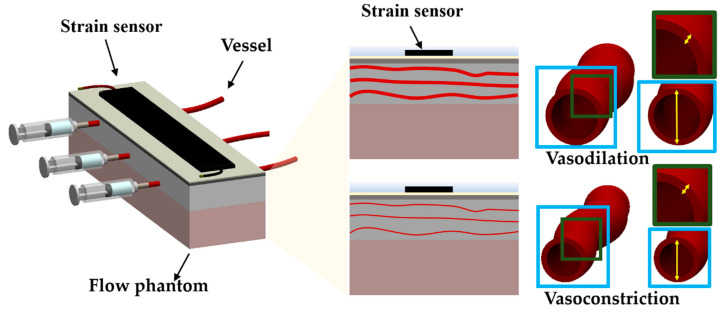
Conceptual illustration of measuring vessel diameter changes using the epidermal strain sensor. Blue boxes in the picture represent cross sections of vessels. Green boxes represent one quarter of the cross sections of the vessels. Yellow arrows in the blue boxes represent the diameters of the vessels. White arrows in the green boxes indicate the thickness of the vessels.

**Figure 2 sensors-21-02201-f002:**
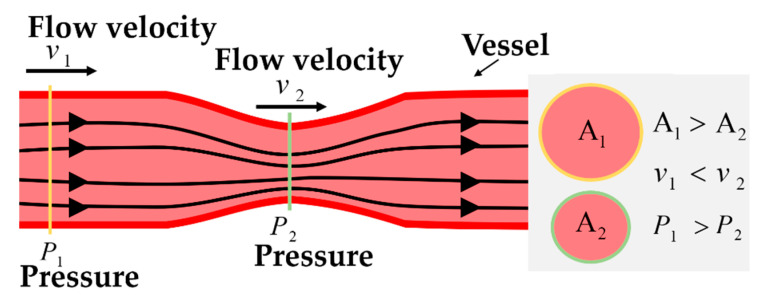
Bernoulli’s law in the vessel.

**Figure 3 sensors-21-02201-f003:**
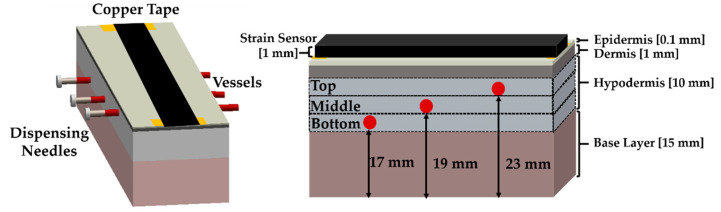
Structure of the simple flow phantom.

**Figure 4 sensors-21-02201-f004:**
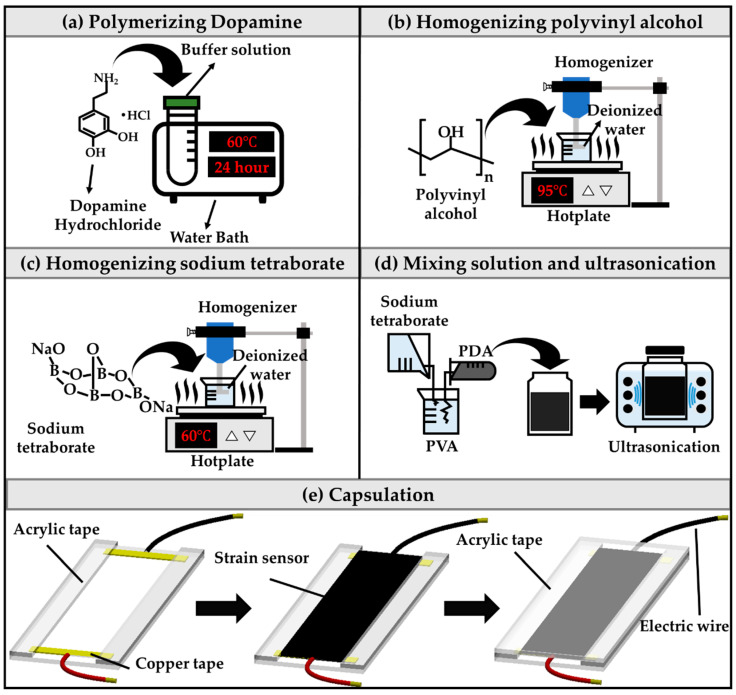
Complete process of sensor fabrication. (**a**) Dopamine was polymerized in a buffer solution (pH = 8) at 60 °C for 24 h to obtain (PDA). (**b**) Polyvinyl alcohol (PVA) solution was prepared by mixing PVA powder with deionized water at 95 °C using a homogenizer. (**c**) Sodium tetraborate powder and deionized water were mixed to form sodium tetraborate solution at 60 °C using the homogenizer. (**d**) PDA, PVA, and sodium tetraborate solution were mixed in a sealed glass bottle for ultrasonication. (**e**) Illustration showing the acrylic tape encapsulating the strain sensor.

**Figure 5 sensors-21-02201-f005:**
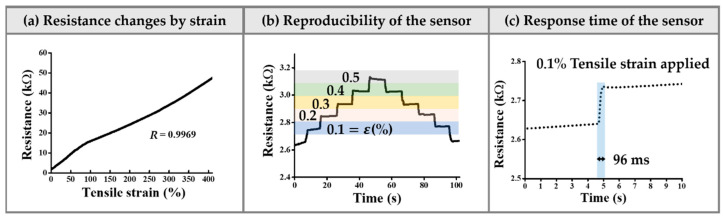
Sensing properties of the strain sensor. (**a**) Changes in electrical resistance for 0% to 450% tensile strain (R = 0.9969). (**b**) Resistance of the strain sensor with 0.1% step-keep changes in strain. (**c**) Sensor response time at 0.1% tensile strain.

**Figure 6 sensors-21-02201-f006:**
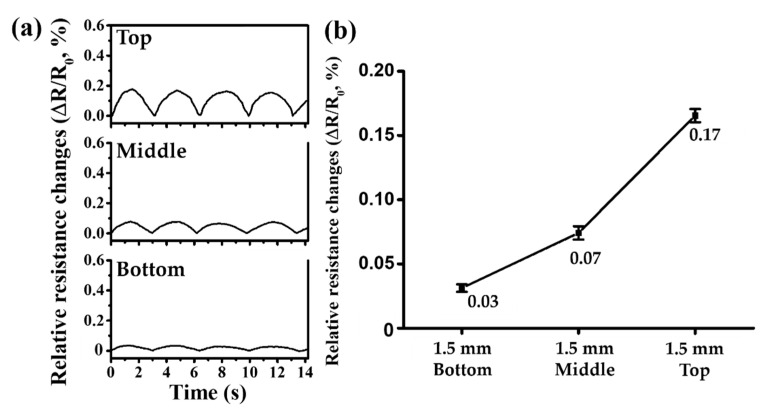
Experimental results of the relative electrical resistance changes in the vessels of 1.5 mm diameter: (**a**) changes in relative electrical resistances at the Top, Middle, and Bottom positions; (**b**) graph of mean relative changes in electrical resistances at the Top, Middle, and Bottom positions.

**Figure 7 sensors-21-02201-f007:**
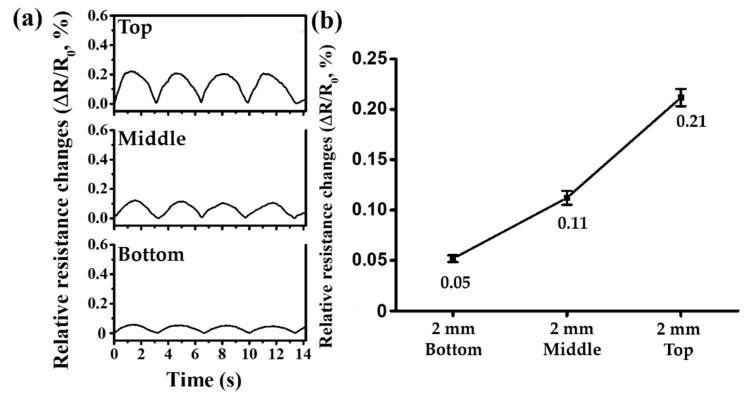
Experimental results of the relative electrical resistance changes in the vessels of 2 mm diameter: (**a**) changes in relative electrical resistances at the Top, Middle, and Bottom positions; (**b**) graph of mean relative changes in electrical resistances at the Top, Middle, and Bottom positions.

**Figure 8 sensors-21-02201-f008:**
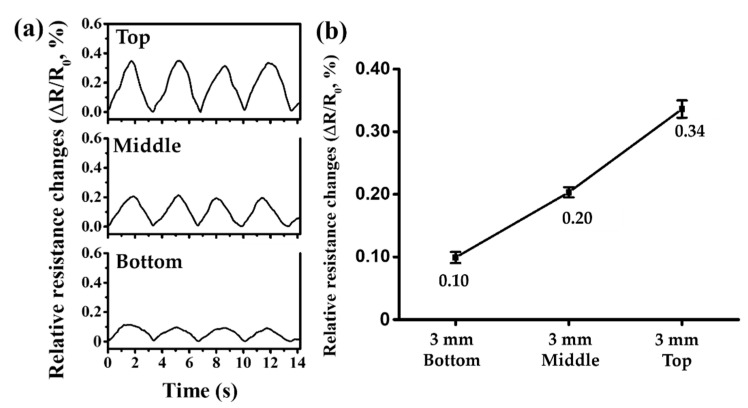
Experimental results of the relative electrical resistance changes in the vessels of 3 mm diameter: (**a**) changes in relative electrical resistances at the Top, Middle, and Bottom positions; (**b**) graph of mean relative changes in electrical resistances at the Top, Middle, and Bottom positions.

**Table 1 sensors-21-02201-t001:** Comparison of methods to monitor blood flow.

Type	Surgical Process	Cost	Trained Operator	Characteristics
Doppler flowmetry	Unnecessary	High	Needed	High accuracy Complicated operation
Spectroscopy	Unnecessary	High	Needed	High accuracy Complicated operation
Computed tomography	Unnecessary	High	Needed	High accuracy Complicated operation
Implantable sensor	Needed	Low	Unnecessary	High accuracy
Fiber Bragg Grating sensor	Unnecessary	Low	Unnecessary	Need calculation
Epidermal sensor	Unnecessary	Low	Unnecessary	Easy operation

**Table 2 sensors-21-02201-t002:** Composition of the flow phantom.

Phantom Layer	Epidermis	Dermis	Hypodermis	Base Layer
Gelatin (wt%)	5	24	2	8
Agar (wt%)	-	1	0.2	-
Other	5 wt% glycerol	1 wt% 1,2-hexanediol	1 wt% 1,2-hexanediol, Vessel (single layer)	1 wt% 1,2-hexanediol

**Table 3 sensors-21-02201-t003:** Young’s modulus for each layer of the simple flow phantom and human skin.

Skin Layer	Epidermis	Dermis	Hypodermis	Reference
Young’s Modulus @ Phantom (kPa)	850–1100 (ε<0.2) $\left(\dot{\varepsilon }=10^{-1}s^{-1}\right)$	47–53 (ε<0.1) $\;\left(\dot{\varepsilon }=10^{-1}s^{-1}\right)$	1.38–2.13 (ε<0.1) $\;\left(\dot{\varepsilon }=10^{-1}s^{-1}\right)$	[26]
Young’s Modulus @ Human (kPa)	1000 (forearm)	56 (forearm)	2 (forearm)	[27,28,29]

## Data Availability

Not applicable.

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
