# Peer review of "Noninvasive Flow Monitoring in Simple Flow Phantom Using Resistive Strain Sensors"

_sensors, 2021, doi:10.3390/s21062201_

Round 1

Reviewer 1 Report

Reviewer’s comments:

The paper experimentally revealed a novel noninvasive resistive strain sensor to monitor the vasodilation and vasoconstriction from skin deformations by measuring the flow of vessel on epidermis for three types of vessel diameters (1.5 mm, 2 mm, and 3 mm) using a simple flow phantom. Also the simple flow phantom to module the human skin was experimentally completed.

  1. The response time has been presented in the report. However, the sensitivity of resistive strain sensor in the measurement of vessel should be exhibited by calculating using Fig. 5 and compared with other ever reported researching results or simulating results.
  2. The experimental results of research in the change of resistance value as the function of tensile strain using resistive strain sensor how to match to the actual human vasodilation and vasoconstriction values should be explored in detail.

Author Response

Dear Editor,

Thank you for allowing resubmission of our manuscript to address reviewer's comments. In the revised manuscript, we thoroughly discussed all issues that reviewers raised, making our manuscript end in much better shape. We believe that the manuscript is now ready for publication in SENSORS.

We are uploading (a) our point-by-point response to the comments of the Reviewers 1~4, (b) an updated manuscript with red highlighting indicating changes, and (c) a clean updated manuscript without highlights (PDF main document).

Best regards,

Dr. Dae Yu Kim 

Associate Professor

Electrical Engineering, College of Engineering, Inha University

Incheon 22212, Republic of Korea

Phone) +82-32-860-7394

E-Mail) [email protected]

Reviewer 2 Report

In this study, a monitoring method for flow expansion and contraction in a simple flow phantom by electrical resistance changes in an epidermal strain sensor attached to the flow phantom. Characteristics was introduced.

The research incudes both numerical analysis and experimental work. The obtained results are promising. In addition, the paper’s subject could be interesting for readers of Sensors journal. Therefore, I recommend this paper for publication in this journal but before that, I have a few comments on the text that should be addressed before publication:

Comments:

1)The novelty of current research has not been explained in introduction section. I recommend the authors to add an explanation at the end of introduction section about novelty of current research in comparison with previous published papers that used electrical resistance for monitoring flow.

2) I highly recommend the authors to add some references in introduction section about other techniques used for measuring other fluid flow characteristics. I recommend the others to add all the following references, which are the newest references in this field, somewhere in the introduction section. In fact, the references refer to the most typical techniques (electrical capacitance [1], photon attenuation [2] and ultrasound [3]) for determining other fluid flow characteristics such as void fraction, flow regime and flow rate.

In recent years, in addition to Venturi he authors are remore new references in the paper in the mentioned field. Some suitable and new references are listed in the following:

[1] Xu, Z., Wu, F., Yang, X. and Li, Y., 2020. Measurement of gas-oil two-phase flow patterns by using CNN algorithm based on dual ECT sensors with venturi tube. Sensors, 20(4), p.1200.

[2] Roshani, M.,, et al., 2021. Evaluation of flow pattern recognition and void fraction measurement in two phase flow independent of oil pipeline’s scale layer thickness. Alexandria Engineering Journal.

[3] Fang, L., Zeng, Q., Wang, F., Faraj, Y., Zhao, Y., Lang, Y. and Wei, Z., 2020. Identification of two-phase flow regime using ultrasonic phased array. Flow Measurement and Instrumentation, 72, p.101726.

3)If it is possible, please add error bars to the data shown in plots of figure 5.

Author Response

(The authors gave the same response as above.)

Reviewer 3 Report

1) Figure 3 is quite gross:

- The black thick arrows are used to sign the layers, as well as heights of 17 mm, 19 mm, and 23 mm from the bottom of the container. It is necessary to designate them otherwise. It is better to do this using a classic style of marking the dimensions in the drawings (the thickness of the layers, and the distance from the bottom of the phantom to the center of the vessels at different levels).

- Each vessel diameters should be designated in different ways, as there are three diameters described.

- It is not clear that there are 4 layers in the left part of the figure – base, hypoderma, dermis and epidermis.

- It is necessary to paint three layers (top-mid-bot) in the right part of the figure for better understanding.

2) Line 130: It is not clear how many vessels were there: three vessels with three diameters or three vessels of different diameters in each layer? It is important to write in the text.

3) The phantom’s manufacturing was descripted in detail. In lines 148 and 149, it is better to write specifically about the consistent immersion of the tubes of each diameter in a separate layer and not to use the term "The flow phantom" for their designation, as it describes the developed device in general.

4) Line 154: Why is it necessary to use 3000 RPM for homogenization at the third stage?

5) Line 155: "The prepared solution is then cooled to 40 °C and added over the hypodermis layer to a thickness of 1 mm."

There must be a mistake here. I suppose this is the 1 mm layer of dermis.

6) Line 174: "The sensor using PDA has self-adhesion characteristics, which attaches well to human as well as porcine skin (gelatin)."

What is the purpose of providing this information? Сould you please provide the references?

7) Line 189: "When the process of ultrasonication is complete, the epidermal sensor is obtained."

At this stage, the sensor is not obtained yet. This is the process of manufacturing the base for the sensor (copper tape). This is the process of manufacturing the base layer for the immersion of the strain sensor.

8) The article contains a detailed description of the sensor, which requires a description of its main part – the parameters of the copper wire (diameter, length, laying method in the 3M tape).

9) Line 209: The article describes the simulation of vasodilation and vasoconstriction processes, however the Young's modulus of polyvinyl alcohol tubes is not listed.

10) Line 213: "The larger the diameter of the vessel, the greater is the pressure exerted on the flow phantom."

The statement is not clear. Was applying the pressure on the skin meant here?

11) Line 216: How was the constant water temperature maintained?

12) Line 216: "For adults, the vessels on the face are usually 1-3.6 mm in diameter [10, 35], carotid vessels are mainly 2-3 mm in diameter [36], and blood vessels on the arms are less than 4 mm in diameter [37, 38]; thus, we choose the diameter of these representative vessels in this experiment as 1.5 mm, 2 mm, and 3 mm, respectively."

It is important to transfer these details to the description of sensor characterization. It is necessary to indicate that the phantom imitates quite large vessels.

13) It is unfortunate that there is no illustration for the final representation of sensor location on the phantom. How does it cover all the vessels? How evenly does the stretching of copper wire occur due to changes in pressure in vessels of different diameters? It would be interesting to see the actual photos of the phantom.

14)         Fig. 5

Response time – sometimes called the sensor time constant – is the time, in seconds, required for a sensor signal to change from 0 to 63.2% of the full scale when the strain sensor is exposed to an instantaneous full scale strain change.

Does the value in the figure correspond to this definition? Was another level selected to be reached?

15)         Fig. 5. a, b.

Why is the different initial electrical resistance from the sensor without deformation? In the text (line 239) is written from 1.8 kΩ.

16) Fig. 6.

The number of decimal places is not justified. There are three significant numbers. For the number 0, it is better no to use 0.0, 0.000 record format. Throughout the text, you need to correctly specify significant numbers.

17) For all measurements, it is not indicated how the statistics were collected. What measurements were used to calculate the standard deviations?

18) Lines 345-347: "In the future, our research will focus on these issues to improve the reliability of the 347 vessel monitoring method for clinical applications."

The sentence was repeated twice.

19)         Line 359: "We, therefore, suggest the proposed method to easily identify the relative diameters and positions of vessels."

This statement is controversial. Even this paper demonstrated that the sensor has the similar response for Top (2 mm) and Mid (3 mm) (Fig. 7 a and Fig. 8 a).

Author Response

(The authors gave the same response as above.)

Reviewer 4 Report

The authors have reasonably addressed the comments and taken into the proper responses. 

Author Response

(The authors gave the same response as above.)

Round 2

Reviewer 2 Report

All the required modifications have been correctly addressed. I believe the manuscript in present form is suitable for publication. Congratulation on authors!

Author Response

We appreciate your time and consideration for reviewing our manuscript.

Reviewer 3 Report

The authors changed the manuscript satisfactorily. The article is possible to publish. However, I would recommend rounding the numbers in Figures 6-8 to the second decimal place.

Author Response

We appreciate your time and consideration for reviewing our manuscript. We rounded to the second decimal place and modified the decimal places of Fig6, 7, and 8 as well as text in our manuscript.

This manuscript is a resubmission of an earlier submission. The following is a list of the peer review reports and author responses from that submission.

Round 1

Reviewer 1 Report

In this paper, a non-invasive vascular detection method using resistance strain sensor is proposed. With the help of the corresponding hydrodynamic formula, the author has established a theoretical model of hemodynamics.

Furthermore, the authors build a skin vascular model, combined with the strain sensor to verify the expected theoretical results. These works can be helpful for cardiovascular research. However, there are some weaknesses and mistakes.

1, The authors did not mention the relevant parameters of the blood vessel in the model, including the thickness of the vessel wall, the material composition of the vessel, etc. According to formula (6), the thickness of the vessel wall directly affects the circumferential stress. In addition, the material composition of the vessel is not given in Table 1. These two parameters have great influence on the experimental results and should be specified in detail.

2, The authors used gelatin, agar and other materials to construct the skin vascular model, and the basis for selecting these materials and composition proportion should be given. If it is based on Young's modulus, Young's modulus of real skin vessels and Young's modulus of the mixed materials used here should be given. Similarly, the source of the size parameters in the model in Figure 3 should also be given, such as from some medical databases or some references. These mechanical parameters and size parameters directly affect the experimental results.

3, According to the experimental results in figures 6 to 8, it can be seen that increasing the diameter of the blood vessel or decreasing the distance of the blood vessel to the body surface will increase the relative resistance change of the strain sensor. In real blood vessel measurement, the diameter and position of blood vessels are unknown. How to decouple these influences and how to distinguish the output difference between the two types (thick and deep, and thin and shallow) of vessels.

4, The hypothesis of the vascular model is too simple. The real blood circulation system is a closed circuit; from aorta to capillaries, blood vessels pass through layers of branches, the diameter is gradually reduced, the flow resistance is increased; there are multiple blood reflection points in the periphery. These phenomena can affect the final experimental results. Although blood can be simplified as laminar Newtonian fluid, it is too simplistic to simulate blood vessels with only a 120 mm elastic tube with both ends open.

In previous studies, such as the literature https://doi.org/10.1016/S1350-4533 (99) 00039-9, researchers used closed nonlinear elastic tube to simulate blood vessel, and added flow resistor, windkessel and reservoir to the circuit, which was much closer to the real situation of blood vessel.

5, Wrong parameters used to simulate pulse. There are two main errors. One is the frequency of the pulse, the other is the flow of the pulse. “The time between pulses is approximately 3.32 s.” (lines 269-270). While the cardiac cycle is about 0.8 s, and are between 0.5 and 1.2 s for most people. “We regularly changed the pumping rates of 6.25 ml/min and 10.25 ml/min using a programmable sequencing pump (NE 4000, new era Inc., USA) to create fluid flow.” (lines 206~208). In this paper, three kinds of vascular models with diameter of 1.5 mm, 2 mm and 3 mm are used. According to the maximum flow rate Q= 10.25 ml/min and the thinnest vessel d =1.5 mm, the maximum velocity of blood flow should be v = Q / S ≈ 0.097m/s, which is much lower than the normal pulse wave velocity (4 ~ 10 m/s). In fact, the stroke volume of the left ventricle is about 70 ml, and the pumping volume of the heart per minute is about 5000 ml at a normal heart rate of 75 bpm, which is much higher than the author's setting of 10.25 ml / min. Therefore, compared with the author's experimental conditions, the real pulse has higher frequency and higher speed. Whether the sensor measurement results in the real situation can be consistent with the expected theoretical conjecture?

6, Using the Bernoulli's law to establish the theoretical model is in doubt (corresponding to the formula (4), (5) and Figure 2 in the paper). The premise for the application of the Bernoulli's law is energy conservation. Referring to Fig. 2, Bernoulli's law describes the process of the same flow (i.e. the flow rate is consistent), and the difference of different thicknesses in the same pipe. In the real blood flow, the energy of the vascular system is not conserved when the heart ejects blood and works externally. For the diastolic and systolic blood vessels, it is equivalent to two different flow processes (i.e. the flow rate is inconsistent). In this case, can Bernoulli's law still be applicable?

7, The skin strain sensor used in this paper is the same with the previous works. DOI: 10.1021/acssuschemeng.9b00579;DOI: 10.1002/marc.201900450;DOI: 10.1039/c8tc00157j。

8, Some obvious mistakes must be caused by clerical error. “high sensitivity to measurements of small modulus (1

Reviewer 2 Report

The paper measures changes in resistance of a strain sensor placed on the surface of a tissue phantom. Different configurations are investigated

It’s not really clear what you would like to obtain from the measurement and this needs to be explained more explicitly in the introduction. Heart rate is clearly achievable but anything else will be much more difficult. For example eq 6 shows that the measurement depends on vessel diameter, wall thickness and pressure. If one adds vessel depth as a variable then it is an ill-conditioned problem to extract anything other than heart rate.

The novelty needs to be more explicit in the introduction, the discussion lines 325-330 says that strain sensors have been used before and mathematical models already developed so what is new in this paper.

Line 17 - <1%

Line 25 – This study presents a method for noninvasive vessel monitoring, this needs to be softened a little as this has not been demonstrated, it may have potential in future.

Line 52 – blood pressure has also been measured using fibre bragg gratings e.g. https://doi.org/10.3390/s19235088 . It would be useful to include this technology and discuss the relative merits of the different approaches, using a table if appropriate. You might also mention PPG for heart rate measurements depending on your main objective.

Equation 6 – how do you decouple the 3 parameters if you only measure hoop stress, depends on thickness of vessel, vessel wall and pressure.

Table 1 – you have stated that the mechanical properties of the phantom are similar to those of tissue but don’t state Young’s modulus or Shear modulus in the equation. Are you able to measure or calculate and add these to the table, making a comparison to tissue values from the literature.

L155 – epidermal typo

Discussion - needs to discuss whether the the problem is well conditioned. The results demonstrate it is not as the same values are obtained for some different vessel thicknesses and depths.

Discuss the affect of other factors such as stiffness of the vessel or skin hydration have on the measurement. More detail needed than just stating future work

Reviewer 3 Report

This paper introduces an epidermal strain sensor for monitoring vascular expansion and contraction in the skin by electrical resistance changes. An easy and simple way of making skin-vessel phantom as a testbed will be attractive to researchers in the biomedical engineering field. The data from experimentally validated using resistance changes via the heartbeats will provide strong references as well. Also, different dimensions of vessels were adequately evaluated with the Bernoulli equation and hoop stress formula. The paper was well organized and written concisely and smoothly. I would recommend this paper for publication after fixing a few minor English errors as provided in the PDF file.

Round 2

Reviewer 1 Report

The authors answered the questions one by one, and basically solved the questions about the sources and citations of the parameter data used in the experiment. However,  for the key problems related to the experimental process and principle, the replies are unconvincing. “Our experiments are focused on presenting a novel method for monitoring vessel movement using fabricated strain sensors” is emphasized in the reply. However, the fact is that the scheme proposed by the author can not solve the problem of multi sensitive source decoupling, and the feasibility of the method is still in doubt. In addition, the experimental verification scheme proposed by the author is quite different from the actual cardiovascular situation, which can not explain the reliability and practicability of the experimental verification, and the conclusion of the article lacks sufficient proof. The revision is not sufficient. The main content of the article has not been improved. Therefore, I do not think it is acceptable.

Reviewer 2 Report

The authors have addressed my comments, most of these are addressed by highlighting weaknesses and suggesting future work but this makes the paper suitable for publication.